

# Data concordance between ESRD Medical Evidence Report and Medicare claims: is there any improvement?

Yi Mu[1], Andrew I. Chin[2,3], Abhijit V. Kshirsagar[4] and Heejung Bang[5,6]

[1] Office of Population Health and Accountable Care, UCSF Medical Center, University of California, San Francisco, San Francisco, CA, United States of America

[2] Division of Nephrology, University of California, Davis School of Medicine, University of California, Davis, Sacramento, CA, United States of America

[3] Division of Nephrology, Sacramento VA Medical Center, VA Northern California Health Care Systems, Mather Field, CA, United States of America

[4] UNC Kidney Center and Division of Nephrology and Hypertension, University of North Carolina at Chapel Hill, Chapel Hill, NC, United States of America

[5] Division of Biostatistics, Department of Public Health Sciences, University of California, Davis, Davis, CA United States of America

[6] Center for Healthcare Policy and Research, Davis School of Medicine, University of California, Davis, Sacramento, CA, United States of America

Corresponding author
Heejung Bang, hbang@ucdavis.edu

## ABSTRACT

**Background**. Medicare is one of the world's largest health insurance programs. It provides health insurance to nearly 44 million beneficiaries whose entitlements are based on age, disability, or end-stage renal disease (ESRD). Data of these ESRD beneficiaries are collected in the US Renal Data System (USRDS), which includes comorbidity information entered at the time of dialysis initiation (medical evidence data), and are used to shape health care policy. One limitation of USRDS data is the lack of validation of these medical evidence comorbidities against other comorbidity data sources, such as medical claims data.

**Methods**. We examined the potential for discordance between USRDS Medical Evidence and medical claims data for 11 comorbid conditions amongst Medicare beneficiaries in 2011–2013 via sensitivity, specificity, kappa and hierarchical logistic regression.

**Results**. Among 61,280 patients, most comorbid conditions recorded on the Medical Evidence forms showed high specificity ($>0.9$), compared to prior medical claims as reference standard. However, both sensitivity and kappa statistics varied greatly and tended to be low (most $<0.5$). Only diabetes appeared accurate, whereas tobacco use and drug dependence showed the poorest quality (sensitivity and kappa $<0.1$). Institutionalization and patient region of residency were associated with data discordance for six and five comorbidities out of 11, respectively, after conservative adjustment of multiple testing. Discordance appeared to be non-informative for congestive heart failure but was most varied for drug dependence.

**Conclusions**. We conclude that there is no improvement in comorbidity data quality in incident ESRD patients over the last two decades. Since these data are used in case-mix adjustment for outcome and quality of care metrics, the findings in this study should press regulators to implement measures to improve the accuracy of comorbidity data collection.

## INTRODUCTION

The United States Renal Data System (USRDS) is a national data system that includes extensive information about chronic kidney disease and end-stage renal disease (ESRD) in the US (*USRDS, 2016*). In addition to patient-level data, USRDS also includes specifics of the dialysis clinics in which patients initiate dialysis. In the US, dialysis clinic oversight is divided amongst 18 networks that represent geographic regions of the country. Although the main strengths of the USRDS are its size and representativeness, its limitations include lack of validated comorbidity information and lack of complete laboratory data at initiation of renal replacement therapy (*Foley & Collins, 2013*).

The USRDS provides two major data sources for ascertaining comorbid conditions in patients on dialysis that are being used in current health care policy research and practice. One source is medical claims data and the other is medical evidence data (Medical Evidence Record Form, designated as CMS-2728 by Medicare), a single form completed by health care providers or staff *exclusively* at the time of dialysis initiation, a critical transition period in a patient's life, and providing baseline data upon patient entry into the ESRD program. Although the concise comorbidity data from CMS-2728 are easy to access, there has been ongoing concern about the reliability of CMS-2728 (*Byrne & Vernon, 1991*; *Kim et al., 2012*; *Krishnan et al., 2015*; *Longenecker et al., 2000*; *Solid et al., 2014*).

An important use of comorbidity data in health policy is in case-mix adjustment in the development of quality metrics of individual dialysis clinics, which are used to profile health care providers and facilities (*Ash et al., 2012*). Medicare administers the ESRD Quality Incentive Program (QIP) to promote high quality services in dialysis facilities treating outpatients with ESRD. ESRD-QIP in 2017 included standardized readmission ratio (SRR) as a clinical measure (*CMS, 2016*; *CMS, 2017c*). CMS also developed the Dialysis Facility Compare, a 5-Star Rating system of dialysis facilities for public reporting, which initially included a Standardized Mortality Ratio (SMR) and Standardized Hospitalization Ratio (SHR). SMR and SHR use CMS-2728, while the SRR is calculated using past year claims for each index hospital discharge to adjust comorbidities (*CMS, 2014a*; *CMS, 2017a*; *Kshirsagar et al., 2017*; *UM-KECC, 2017*).

Differences in health care delivery based on local and regional factors, items beyond patient or provider control, are also topics that have been highlighted over the last several years (*Bernheim et al., 2016*; *Kshirsagar et al., 2017*; *Manickam et al., 2017*; *Martsolf et al., 2016a*; *Martsolf et al., 2016b*). For example, regional differences have been associated with outcomes in the ESRD setting, particularly when the West coast is compared against other regions of the US (*Almachraki et al., 2016*; *Kshirsagar et al., 2017*; *Mu et al., 2018*), despite near universal care and coverage offered in the US for ESRD.

Building upon the prior studies, we estimated and compared the prevalence and concordance of 11 common comorbid conditions recorded on CMS-2728 versus those determined by medical claims (Aim 1). We implemented different methods of claims data

processing in recognition that different algorithms are currently being used in different medical conditions—for example, ESRD vs. cancer—and that substantial discordance has been reported in claims even within a one-year period in a given dialysis patient cohort (*Krishnan et al., 2015*). We also examined if associations with regions and other external factors (we call 'environment-related factors') were found in comorbidity ascertainment between CMS-2728 and medical claims data (Aim 2); if no association is found, we can perhaps be more comfortable in the accuracy of these data (*Solid et al., 2014*). Through these analyses, we intended to study if there has been any improvement in comorbidity ascertainment data quality since 2010.

## MATERIALS AND METHODS

### Study design and data sources

Our study required comorbidity data prior to and at the time of ESRD. Therefore, we chose to examine subjects who had Medicare parts A and B coverage and medical claims information prior to ESRD. Since Medicare eligibility begins at age 65, we looked at those who started dialysis at age ≥67 years of age, allowing for *at least* one year of claims data prior to ESRD, though this age restriction cannot guarantee the completeness of the past two years of Medicare claims data.

Specifically, we included Medicare-eligible patients in the USRDS ≥67 years of age at the time of dialysis initiation, who were also enrolled in Medicare prior to ESRD, and who started hemodialysis between January 1, 2011 and June 30, 2013. Additional inclusion criteria included subjects who had a completed CMS-2728 form and had Medicare parts A and B as primary payer 1 year prior to and at time of dialysis initiation. The final study cohort included 61,280 patients; see Fig. 1. To ensure Medicare enrollment, we linked the USRDS to pre-ESRD Medicare claims data. We next linked the USRDS to the Dartmouth Atlas of Health Care and the American Community Survey to provide environment-related variables.

### Comorbidity and ascertainment algorithms

We included 11 comorbid conditions found on the CMS-2728 form: atherosclerotic heart disease (AHD), congestive heart failure (CHF), cerebrovascular disease (CBVD), peripheral vascular disease (PVD), other cardiac diseases, chronic obstructive pulmonary disease (COPD), cancer, diabetes mellitus (DM), alcohol dependence, drug dependence, and tobacco use. To ascertain comorbid conditions using past-year Medicare claims, we considered the following three methods for data processing:

- Method A: We required at least one claim from an inpatient/home health agency/skilled nursing facility/hospice *or* at least two claims from an outpatient/physician-supplier.
- Method B: We required at least two claims from outpatient/physician-supplier more than 30 days apart, in addition to the criteria in Method A.
- Method C: We excluded co-existing radiology, diagnostic laboratory, and durable medical equipment physician-supplier. We then applied Method A.

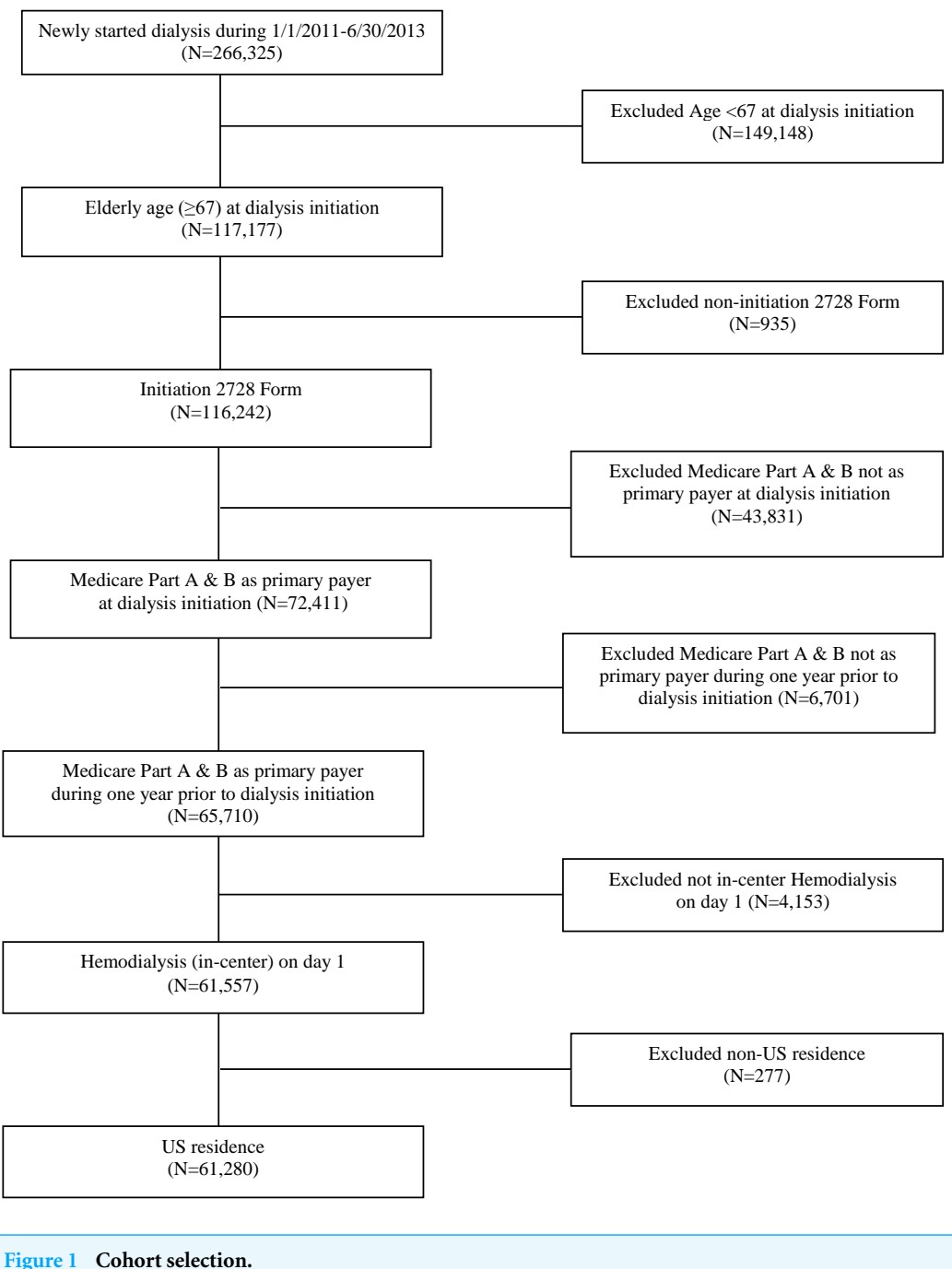

**Figure 1 Cohort selection.**

Method A may be considered the current norm found in nephrology health care research (*Krishnan et al., 2015*), while Methods B and C were informed by cancer research, such as the Surveillance, Epidemiology, and End Results (SEER) program—using the SEER-Medicare dataset (*Baldwin et al., 2006*). We considered different processing methods because substantial discordance was found between claims ascertained before and shortly after dialysis initiation in a prior study on this subject matter (*Krishnan et al., 2015*).

## Environment-related factors

We considered the external, regional and geographic factors as potentially influencing the outcome, data discordance (*Almachraki et al., 2016*; *Kshirsagar et al., 2017*; *Mu et al., 2018*). Specifically, we considered the following seven factors: (1) patient's institutionalization status; (2) health care utilization intensity in patient's residence; (3) dialysis clinic size (patient volume); (4) geographic location/region of the country of the dialysis facility; (5) dialysis facility's rural–urban commuting area (RUCA); (6) regional poverty level; and (7) regional education level. As the study by *Krishnan et al. (2015)* had centered upon patient-level factors (e.g., demographics), we decided to include these facility/area-level factors. The institutionalization status of patients was defined as nursing home, assisted living, or other. Dialysis facility geographic location was assigned to 1 of 4 regions (Northeast, South, Midwest and West), which was determined by the network in which the clinic resided (*Dalrymple et al., 2014*; *Kshirsagar et al., 2017*). Dialysis facility volume reflected the number of patients receiving hemodialysis in a year. To assign dialysis facilities to a rural or urban area, we used ZIP code level RUCA (*RUCA, 2007*) and ZIP code tabulation area (ZCTA) through the Uniform Data System Mapper (http://udsmapper.org/zcta-crosswalk.cfm), then linked it to the 2010–2014 American Community Survey (https://www.census.gov/programs-surveys/acs/). We selected two regional socio-economic indicators, both at the ZCTA level: percent of population below the federally defined poverty line, and percent of population amongst persons aged ≥25 years who have at least a high school education (*Manickam et al., 2017*). We linked the USRDS to the 2011–2013 Dartmouth Atlas of Health Care (http://www.dartmouthatlas.org), and selected a measure of intensity of health care utilization—reimbursements per decedent for inpatient hospitalizations during the last six months of life—and matched it to the patient's state of residence and year of dialysis initiation (*Song et al., 2010*).

## Statistical analysis

Descriptive statistics were used to describe cohort characteristics and comorbidity prevalence. For example, standard diagnostic statistics (sensitivity (SN), specificity (SP)) and concordance statistic (kappa) were computed to assess the agreement of the two data sources for each comorbid condition, along with McNemar's test for correlated data; a larger value close to 1 is interpreted as higher agreement for SN, SP and kappa although kappa can be low for a variable with low prevalence despite high agreement (*Byrt, Bishop & Carlin, 1993*). We chose >0.5 as an ad-hoc threshold for acceptability. We intentionally did not attempt to assess accuracy because neither data source can be considered as gold standard. To examine factors that are associated with the outcome (discordance = 0/1

**Table 1** Cohort characteristics ($N = 61,280$).

| Patient level | Category | N | % | Facility level | Category | N | % |
|---|---|---|---|---|---|---|---|
| Age | [67,75) | 24,336 | 39.7 | Number of patients per facility (volume) | Missing | 899 | 1.5 |
| | [75,85) | 27,616 | 45.1 | | ≤40 | 10,000 | 16.3 |
| | ≥85 | 9328 | 15.2 | | 41–63 | 13,651 | 22.3 |
| Gender | Female | 28,088 | 45.8 | | 64-91 | 16,670 | 27.2 |
| | Male | 33,190 | 54.2 | | >91 | 20,060 | 32.7 |
| | Unknown | 2 | | Region | Missing | 583 | 1 |
| Race | Black | 11,514 | 18.8 | | Northeast | 11,227 | 18.3 |
| | White | 46,956 | 76.6 | | South | 23,857 | 38.9 |
| | Other | 2,810 | 4.6 | | Midwest | 15,508 | 25.3 |
| Ethnicity | Non-Hispanic | 56,417 | 92.1 | | West | 10,105 | 16.5 |
| | Hispanic | 4,863 | 7.9 | RUCA | Missing | 750 | 1.2 |
| Primary cause of ESRD | Diabetes | 24,834 | 40.5 | | Urban | 47,649 | 77.8 |
| | Hypertension | 23,098 | 37.7 | | Large Rural | 8,865 | 14.5 |
| | Glomerulo-nephritis | 2,806 | 4.6 | | Small Rural | 3,267 | 5.3 |
| | Other | 10,542 | 17.2 | | Isolated Small Rural | 749 | 1.2 |
| Healthcare Utilization[a] | 1 | 5,168 | 8.4 | All people below poverty in past 12 months, % | Missing | 719 | 1.2 |
| Quartile | 2 | 11,456 | 18.7 | | <5 | 3,737 | 6.1 |
| | 3 | 22,175 | 36.2 | | 5–9.9 | 12,048 | 19.7 |
| | 4 | 22,481 | 36.7 | | 10–14.9 | 12,384 | 20.2 |
| Institutionalization | No | 52,165 | 85.1 | | 15–19.9 | 12,407 | 20.3 |
| | Yes | 9,115 | 14.9 | | 20–24.9 | 8,964 | 14.6 |
| Prior nephrology care | <6 months | 8,590 | 14 | | ≥25 | 11,021 | 18 |
| | 6–12 months | 10,569 | 17.3 | Adults ≥25 yr who has Bachelor or higher, % | Missing | 627 | 1 |
| | >12 months | 18,607 | 30.4 | | <20 | 20,931 | 34.2 |
| | No | 15,336 | 25 | | [20,30) | 17,410 | 28.4 |
| | Unknown | 8,178 | 13.4 | | ≥30 | 22,312 | 36.4 |

**Notes.**
[a] Healthcare utilization is defined as reimbursement per decedent for inpatient hospitalization during the last 6 months at state level.

for No/Yes) using Method B, we implemented a hierarchical logistic regression model with dialysis facility as random intercept and multi-variable adjustment (listed in Table 1) (*Fitzmaurice, Laird & Ware, 2011*). We computed conditional odds ratio (OR) along with 95% confidence interval (CI) and *p*-value, where a larger value of OR above 1 indicates more discordance and a lower value of OR below 1 indicates less discordance, equivalently, more concordance.

As education level and poverty level tend to be correlated, we conducted sensitivity analyses with either education or poverty, not both in the model, to handle the collinearity issue. In order to handle the multiple testing issue, we indicated $p < 0.0001$, in addition to $p < 0.05$, where we used $p < 0.0001$ as a conservative threshold for the summarization of

the results and assessing statistical significance. All analyses were conducted using SAS® version 9.4 (SAS Institute, Cary, NC, USA).

## RESULTS

### Cohort characteristics

Our analyses included 61,280 patients from 5,588 dialysis facilities who initiated in-center hemodialysis during the period of 2011-2013 and who were also Medicare beneficiaries during the year prior to dialysis initiation; see Fig. 1 for inclusion/exclusion criteria. Table 1 summarizes the study cohort regarding patient- and facility/area-level characteristics. About 60% of patients were 75 years old or older. About 75% of patients were White, 19% were Black, and 8% of patients identified as Hispanic. The most common primary causes of renal failure were diabetes (41%) and hypertension (38%).

### Comorbidities: prevalence, sensitivity, specificity, and kappa

We compared the prevalence of 11 comorbidities and the agreement of data (presence vs. absence of comorbidity) between CMS-2728 and Medicare claims, where claims were processed by the three different methods described earlier; see Table 2. We used claims data as the reference standard in all comparisons. Among comorbid conditions, the category 'other cardiac diseases' was largely affected by which particular method was used to process the claim (41.8, 32.3, 26.8% with methods A, B, C, respectively). In general, prevalence of comorbidities based on CMS-2728 was uniformly lower than that based on claims data. This discordance was especially notable for some conditions (i.e., AHD, COPD, CBVD, PVD, alcohol dependence, drug dependence, and tobacco usage) where prevalence estimates on the CMS-2728 form were less than half of that which was determined by claims.

Overall specificity was high; most comorbidities showed specificity >0.9, except for AHD (0.86–0.85), CHF (0.79–0.84), and other cardiac diseases (0.77–0.78). However, sensitivity varied dramatically across comorbid conditions, ranging from 0.04 to 0.86. The comorbidity of DM had the highest sensitivity (SN 0.83–0.86), a lower but acceptable value for CHF (SN 0.57–0.59) and all other conditions showing SN<0.5. The lowest sensitivity was observed for tobacco use and drug dependence (SN< 0.1). The corresponding kappa statistics between the two data sources also varied markedly, ranging from 0.07 to 0.73. Once again, the lowest agreement (kappa ≤ 0.1) was observed in drug dependence and tobacco use. DM showed the highest agreement (~0.72), while all other conditions showed <0.5.

In terms of the three methods for claims data processing, Methods B and C tended to yield lower prevalence of each comorbidity, but a slightly higher sensitivity with specificity virtually unchanged, compared to Method A.

### Factors associated with discordance between CMS-2728 and claims

Table 3 presents ORs along with 95% CIs for each factor and discordance for 10 comorbidities, where alcohol dependence was excluded due to non-convergent model fit. Institutionalization and geographic region were significantly associated with data discordance for the largest number of the comorbidities we examined; an institutionalized patient status was associated with discordance for six comorbidities (i.e., AHD, CBVD,

**Table 2  Prevalence and Agreement: Medical evidence (CMS-2728) vs. claims data ($N = 61,280$).**

| Comorbidity | % (CMS-2728) | Method[a] | % (Claims) | Kappa | Sensitivity | Specificity |
|---|---|---|---|---|---|---|
| Diabetes mellitus | 57.3 | A | 64.8 | 0.71 | 0.83 | 0.91 |
| | | B | 62.6 | 0.72 | 0.85 | 0.89 |
| | | C | 61.8 | 0.73 | 0.86 | 0.89 |
| Cancer | 11.9 | A | 18.9 | 0.42 | 0.41 | 0.95 |
| | | B | 15.6 | 0.42 | 0.44 | 0.94 |
| | | C | 15.6 | 0.42 | 0.44 | 0.94 |
| Congestive heart failure | 41.1 | A | 60.7 | 0.38 | 0.57 | 0.84 |
| | | B | 52.8 | 0.38 | 0.59 | 0.79 |
| | | C | 56.0 | 0.39 | 0.59 | 0.82 |
| Chronic obstructive pulmonary disease | 14.3 | A | 33.6 | 0.34 | 0.34 | 0.96 |
| | | B | 30.1 | 0.36 | 0.36 | 0.95 |
| | | C | 31.3 | 0.35 | 0.35 | 0.95 |
| Cerebrovascular disease | 11.6 | A | 23.0 | 0.24 | 0.27 | 0.93 |
| | | B | 17.4 | 0.27 | 0.31 | 0.92 |
| | | C | 17.9 | 0.27 | 0.31 | 0.93 |
| Atherosclerotic heart disease | 27.2 | A | 57.4 | 0.21 | 0.37 | 0.86 |
| | | B | 52.5 | 0.22 | 0.38 | 0.85 |
| | | C | 53.5 | 0.22 | 0.38 | 0.85 |
| Peripheral vascular disease | 16.3 | A | 39.2 | 0.19 | 0.27 | 0.9 |
| | | B | 32.9 | 0.21 | 0.28 | 0.9 |
| | | C | 33.9 | 0.21 | 0.28 | 0.9 |
| Alcohol dependence | 0.8 | A | 2.3 | 0.21 | 0.14 | 1 |
| | | B | 2.0 | 0.2 | 0.15 | 1 |
| | | C | 2.2 | 0.21 | 0.15 | 1 |
| Other cardiac | 26.9 | A | 41.8 | 0.13 | 0.34 | 0.78 |
| | | B | 32.3 | 0.14 | 0.36 | 0.77 |
| | | C | 26.8 | 0.14 | 0.37 | 0.77 |
| Tobacco use | 3.5 | A | 22.5 | 0.1 | 0.09 | 0.98 |
| | | B | 21.4 | 0.1 | 0.09 | 0.98 |
| | | C | 22.1 | 0.1 | 0.09 | 0.98 |
| Drug dependence | 0.1 | A | 1.0 | 0.07 | 0.04 | 1 |
| | | B | 0.9 | 0.07 | 0.04 | 1 |
| | | C | 0.9 | 0.07 | 0.04 | 1 |

**Notes.**

[a] Method A, B, and C: see Method section.

Sensitivity and Specificity were computed with claims data as reference standard. From McNemar's test $p < 0.0001$, except for 'Other cardiac' ($p = 0.60$ for Method C vs CMS-2728).

other cardiac, PVD, COPD, drug dependence) with OR = 1.1–2.3, with $p < 0.0001$. Geographic region of patient residence was associated with 5 comorbidities (i.e., AHD, CBVD, other cardiac, PVD, tobacco use) with OR = 0.5–1.3. In contrast, other factors such as regional indicator of health care utilization, volume/size of the dialysis facility, urbanicity and regional socio-economic indicators (poverty and education levels) did not show systematic associations. Also, discordance appears to be non-informative for CHF

**Table 3  Environment-related factors associated with discordance between CMS-2728 and past year claims data.**

### a. Cardiovascular disease-related comorbidities

| Factors | | Odds ratio (95% confidence interval) | | | | |
|---|---|---|---|---|---|---|
| | | AHD | CHF | CBVD | Other cardiac | PVD |
| Institutionalization | Yes vs. No | **1.1(1.1–1.2)** | 1.1(1.0–1.1) | **1.4(1.3–1.5)** | **1.2(1.2–1.3)** | **1.3(1.3–1.4)** |
| Healthcare | 2 | *1.1(1.0–1.2)* | 1.0(0.9–1.1) | 1.0(0.9–1.1) | 1.0(0.9–1.1) | 1.0(0.9–1.1) |
| utilization | 3 | **1.2(1.1–1.3)** | 1.0(1.0–1.1) | 1.0(0.9–1.1) | 1.0(1.0–1.1) | *1.1(1.0–1.2)* |
| quartile | 4 vs. 1 | **1.4(1.3–1.5)** | 1.1(1.0–1.2) | 1.0(1.0–1.1) | 1.1(1.0–1.1) | *1.2(1.1–1.2)* |
| Volume (no of patients per facility) | 41–63 | 1.0(0.9–1.0) | 1.0(1.0–1.1) | 1.0(1.0–1.1) | 1.1(1.0–1.1) | 1.0(1.0–1.1) |
| | 64-91 | *0.9(0.9–1.0)* | 1.0(0.9–1.0) | 1.0(1.0–1.1) | 1.0(1.0–1.1) | 1.0(1.0–1.1) |
| | >91 vs. ≤40 | *0.9(0.9–1.0)* | 1.0(1.0–1.1) | 1.0(0.9–1.1) | 1.0(1.0–1.1) | 1.0(1.0–1.1) |
| Region | Midwest | **1.2(1.1–1.3)** | 1.0(0.9–1.1) | *1.1(1.0–1.2)* | *1.1(1.0–1.2)* | **1.1(1.1–1.2)** |
| | Northeast | 1.0(1.0–1.1) | 1.0(0.9–1.1) | *1.1(1.0–1.2)* | **1.2(1.1–1.2)** | **1.3(1.2–1.4)** |
| | South vs. West | **1.3(1.2–1.4)** | 1.1(1.0–1.1) | **1.2(1.1–1.3)** | **1.1(1.1–1.2)** | 1.0(1.0–1.1) |
| RUCA | Large rural | **0.9(0.8–0.9)** | *0.9(0.9–1.0)* | *0.9(0.9–1.0)* | 1.0(0.9–1.0) | 1.0(0.9–1.0) |
| | Small rural | 0.9(0.8–1.0) | 0.9(0.8–1.0) | 0.9(0.8–1.0) | 1.0(0.9–1.0) | 0.9(0.8–1.0) |
| | Isolated small rural vs. Urban | *0.8(0.7–0.9)* | 1.0(0.9–1.2) | 1.0(0.8–1.2) | 1.1(1.0–1.3) | 1.1(0.9–1.3) |
| People in poverty, % | <5 | *1.1(1.0–1.3)* | 0.9(0.8–1.0) | *1.1(1.0–1.3)* | *1.2(1.1–1.3)* | 1.1(1.0–1.2) |
| | 5–9.9 | 1.0(1.0–1.1) | 1.0(0.9–1.0) | 1.1(1.0–1.2) | *1.1(1.0–1.2)* | 1.0(1.0–1.1) |
| | 10–14.9 | 1.0(1.0–1.1) | 1.0(0.9–1.0) | 1.1(1.0–1.1) | *1.1(1.1–1.2)* | 1.0(1.0–1.1) |
| | 15–19.9 | 1.0(0.9–1.0) | 1.0(0.9–1.0) | 1.1(1.0–1.1) | *1.1(1.1–1.2)* | 1.0(0.9–1.1) |
| | 20–24.9 vs. ≥25 | 1.0(1.0–1.1) | 1.0(1.0–1.1) | 1.0(1.0–1.2) | *1.1(1.0–1.1)* | 1.0(1.0–1.1) |
| Adults with ≥bachelor degree, % | 20–29.9 | 1.0(0.9–1.0) | 1.0(0.9–1.0) | 1.0(1.0–1.1) | 1.0(1.0–1.1) | 1.0(0.9–1.0) |
| | ≥30 vs. <20 | **0.9(0.8–0.9)** | 1.0(0.9–1.0) | *0.9(0.9–1.0)* | 1.0(0.9–1.0) | *0.9(0.9–1.0)* |

### b. Non-cardiovascular disease-related comorbidities

| Factors | | Odds ratio (95% confidence interval) | | | | |
|---|---|---|---|---|---|---|
| | | COPD | Cancer | Diabetes mellitus | Drug dependence | Tobacco use |
| Institutionalization | Yes vs. No | **1.2(1.2–1.3)** | *0.9(0.9–1.0)* | 1.1(1.0–1.1) | **2.3(2.1–2.6)** | 1.1(1.0–1.1) |
| Healthcare | 2 | 1.0(0.9–1.1) | 1.0(0.9–1.1) | 1.1(1.0–1.2) | 1.1(0.9–1.4) | 1.0(0.9–1.1) |
| utilization | 3 | *1.1(1.0–1.2)* | 1.0(0.9–1.1) | *1.2(1.0–1.3)* | 1.0(0.8–1.3) | 1.0(0.9–1.1) |
| quartile | 4 vs. 1 | *1.1(1.0–1.2)* | 1.0(0.9–1.1) | *1.2(1.1–1.3)* | 1.0(0.8–1.3) | 1.0(0.9–1.1) |

*(continued on next page)*

| Factors | | Odds ratio (95% confidence interval) | | | | |
|---|---|---|---|---|---|---|
| | | COPD | Cancer | Diabetes mellitus | Drug dependence | Tobacco use |
| Volume (no of patients per facility) | 41–63 | 1.0(0.9–1.0) | 1.0(0.9–1.1) | 1.0(0.9–1.0) | **0.7(0.5–0.8)** | 1.0(0.9–1.1) |
| | 64–91 | *0.9(0.9–1.0)* | 1.1(1.0–1.2) | *0.9(0.8–1.0)* | 0.8(0.6–1.0) | 1.1(1.0–1.1) |
| | >91 vs. ≤40 | *0.9(0.9–1.0)* | 1.0(0.9–1.1) | 1.0(0.9–1.1) | 0.8(0.7–1.1) | 1.0(0.9–1.0) |
| Region | Midwest | 1.0(1.0–1.1) | 1.1(1.0–1.2) | 1.0(0.9–1.1) | *0.5(0.4–0.8)* | **1.2(1.1–1.3)** |
| | Northeast | 1.0(1.0–1.1) | *1.1(1.0–1.2)* | *1.2(1.1–1.3)* | 0.6(0.4–0.8) | *1.1(1.0–1.2)* |
| | South vs. West | 1.0(1.0–1.1) | 1.1(1.0–1.2) | *1.1(1.0–1.2)* | *0.7(0.5–0.9)* | 1.0(0.9–1.1) |
| RUCA | Large rural | 1.0(1.0–1.1) | 1.0(0.9–1.0) | 0.9(0.9–1.0) | 0.9(0.7–1.2) | 1.0(0.9–1.1) |
| | Small rural | 1.0(0.9–1.1) | 0.9(0.8–1.0) | **0.8(0.7–0.9)** | *0.4(0.3–0.7)* | 0.9(0.8–1.0) |
| | Isolated small rural vs. Urban | 1.0(0.8–1.2) | 1.0(0.8–1.2) | *0.8(0.6–1.0)* | 1.6(0.8–3.3) | 1.1(0.9–1.4) |
| People in poverty,% | <5 | 1.0(0.9–1.1) | 1.1(1.0–1.3) | 1.0(0.9–1.2) | 0.7(0.4–1.3) | *1.2(1.1–1.4)* |
| | 5–9.9 | 1.0(1.0–1.1) | 1.1(1.0–1.2) | 0.9(0.8–1.0) | *0.6(0.5–0.9)* | *1.2(1.1–1.3)* |
| | 10–14.9 | 1.0(0.9–1.1) | *1.1(1.0–1.2)* | 1.1(0.9–1.1) | 0.8(0.6–1.1) | 1.1(1.0–1.2) |
| | 15–19.9 | 1.0(0.9–1.1) | 1.1(1.0–1.2) | 1.0(0.9–1.1) | 0.8(0.6–1.1) | *1.1(1.0–1.2)* |
| | 20–24.9 vs. ≥25 | 1.0(0.9–1.1) | *1.1(1.0–1.2)* | 1.1(1.0–1.2) | 1.0(0.7–1.3) | 1.1(1.0–1.2) |
| Adults with ≥bachelor degree, % | 20–29.9 | 1.0(0.9–1.0) | 1.0(0.9–1.1) | *0.9(0.9–1.0)* | *1.3(1.0–1.7)* | 1.0(0.9–1.1) |
| | ≥30 vs. <20 | *0.9(0.8–1.0)* | 1.1(1.0–1.1) | *0.9(0.8–1.0)* | 1.3(1.1–1.8) | 1.0(0.9–1.0) |

**Notes.**
Alcohol dependence was not included due to models not convergent.
If $p < 0.0001$, then bold; If $0.0001 < p < 0.05$, then italic, where $p$-values were unadjusted for multiple testing.
AHD, atherosclerotic heart disease; CHF, congestive heart failure; CBVD, cerebrovascular disease; PVD, peripheral vascular disease; COPD, chronic obstructive pulmonary disease; RUCA, rural-urban commuting area.

(OR = 0.9–1.1 with all $p > 0.05$), which may imply good reliability in this variable. In comparison, discordance was most varied for drug dependence (OR = 0.5–2.3).

Finally, our sensitivity analyses that intended to address collinearity between education and poverty showed very similar results and no impact on the key findings reported above, e.g., regarding institutionalization, geographic regions and CHF. Results are summarized in Table S1.

## DISCUSSION

Administrative databases are increasingly used in research, public policy and even in informing health care consumers, as in the case of the Dialysis Facility Compare program. In our study, we utilized two major sources of comorbidity data in nephrology research and health policy in the US—the Medical Evidence report (CMS-2728) that is completed once at the time of dialysis initiation and Medicare claims.

Prior studies on differences in comorbidity ascertainment methodology in ESRD patients have demonstrated a high degree of discordance and overall poor sensitivity, similar to what we found in our study. Longnecker et al. used clinical chart data to validate comorbid conditions reported on CMS-2728 in 1,005 patients who started dialysis from 1995 through 1998. The average sensitivity was 0.59 and specificity was above 0.9, with HIV showing the highest accuracy (*Longenecker et al., 2000*). Krishnan et al. examined comorbidities recorded on CMS-2728 and claims before and after dialysis initiation based on 45,357 Medicare-enrolled patients who initiated dialysis between 2007 and 2009. Excluding DM, the kappa statistic for measuring data concordance (between claims before dialysis initiation vs. CMS-2728 at dialysis initiation, and between claims before vs. after dialysis initiation) ranged from 0.05 to 0.58. They also found that patient demographics and the USRDS network in which the patient received dialysis were associated with data discordance, but did not study specific dialysis facility or regional characteristics, factors that we included in our analysis (*Krishnan et al., 2015*). On the other hand, Solid et al. used Medicare outpatient dialysis claims to validate vascular access—a variable newly added in the 2005 revision of the CMS-2728 form—among patients starting hemodialysis in 2010. The two sources agreed for 94% of 9,777 patients with a kappa statistic of 0.83 (*Solid et al., 2014*). Cause of renal failure and predialysis nephrology care were also evaluated using different data sources over different time periods; the former was reasonably accurate, not the latter (*Byrne & Vernon, 1991*; *Kim et al., 2012*).

The two main goals of our study were (1) to check if there is a meaningful improvement in quality of comorbidity data in the modern era of the many quality-based initiatives, compared to that noted in prior studies; and (2) to identify environment-related (e.g., living condition, socio-economic, dialysis facility or regional area) factors associated with data disagreement. An ancillary goal was to examine different methods of claims data processing, although comorbidity ascertainment algorithm using claims is relatively well established for research purposes, as adapted from the SEER method and Charlson comorbidity index.

Our findings relating to the first goal do not demonstrate a meaningful improvement in data quality. We found similar degrees of poor data discordance and sensitivity between CMS-2728 and medical claims as were noted in prior studies, even in a more contemporary era where the stakes are higher for health care providers of ESRD patients to record accurate comorbidity data. Previous studies reported low kappa and sensitivity, and substantial under-reporting in CMS-2728, and our data did not show that these values have improved over time (*Krishnan et al., 2015*; *Longenecker et al., 2000*). In particular, our study can be directly compared with that by Krishnan et al. with these two studies adopting very similar inclusion and exclusions criteria, using different years of the USRDS. In contrast, we utilized a more contemporary cohort, examined three different methods of claims data processing (including the method of Krishnan et al.), employed multiple statistical measures (beyond kappa statistic), and used a better suited statistical model (accounting for clustering). As anticipated, the two cohorts showed a similar distribution of patient characteristics and similar kappa values. Medicare has tried on their part to improve comorbidity entry by health care providers. As a part of the national effort, CMS provides education and outreach for "Provider Compliance Program" (*CMS, 2014b*).

Regarding the second goal, we newly identified that institutionalization (vs. community-dwelling) and some geographic regions showed a higher frequency of misreporting comorbidity data than others. Midwestern, Northeastern and Southern regions tended to show more data discordance, compared to the Western region in most comorbidities except for drug dependence. Notably, the phenomenon of "West vs. others" in the context of ESRD care has been observed in prior studies with different outcomes (*Kshirsagar et al., 2017*; *Mu et al., 2018*). This regional variation may have policy implications because near universal care and coverage are uniquely available for ESRD by Medicare in the US. Medicare regulators charged with developing future quality metrics that are, in part, based on comorbidities may need to consider these additional regional variations in monitoring comorbidity data reliability. As a model example provided by the investigators from the USRDS data center, 'vascular access type' used for dialysis that is being used for payment (since 2010) successfully passed the data reliability test based on high values in the proportion of agreement as well as kappa statistic, coupled with the absence of statistically significant predictors (among demographics) for disagreement (*Solid et al., 2014*).

We used three different but related methods of claims data processing. While we did not necessarily find one method to be overall superior to the others, we believe that Method A is easiest and most inclusive and Method C may be most restrictive (*Baldwin et al., 2006*; *Krishnan et al., 2015*). It is our opinion that investigator assumptions and preferences, and possibly the particular variable involved, may dictate the method that future investigators choose. We feel that including the three methods in our study might provide some guidance on these decisions. CMS currently established the Medicare and Medicaid Electronic Health Record (EHR) Incentive Programs to encourage health care providers and institutions to adopt, implement, upgrade, and demonstrate meaningful use of certified EHR technology (*CMS, 2017b*). Hopefully with this and advanced medical informatics and algorithms (*Hehner et al., 2017*), comorbidity data capture can be more complete and accurate. Indeed, since the time of this study, Medicare has begun using prevalent comorbidities to the SMR and SHR calculations (*CMS, 2017d*).

Our study has several limitations. First, the findings are limited to elderly in-center hemodialysis patients with Medicare as primary payer prior to dialysis initiation; thus, generalizability to other populations may be limited. Second, neither data source used can serve as a gold standard; thus, we could not assess true data accuracy. Claims data are generally regarded as more accurately recorded, so we used it as the reference standard. Indeed, Krishnan et al. proposed the use of claims during the 3 months after dialysis initiation as a useful source of comorbidity data (*Krishnan et al., 2015*). In our study, kappa (for concordance) and sensitivity (for diagnostics) yielded qualitatively similar results. Third, we did not investigate how certain factors, especially those that are not included in the current profiling model and quality metrics being used by CMS, such as prior nephrology care and geographic region, impact on the profiling status, which is warranted in future research (*Liu et al., 2016*).

## CONCLUSION

Our study raises continued concern about the highly varied data quality in comorbidity information amongst patients with ESRD in the US. Although we used a more recent cohort, we conclude that no improvement was observed over the last two decades, and our results emphasize the concerns about the data accuracy of the comorbidities on the CMS-2728 form and its associated use in developing health policies. Hopefully, these studies provide further incentive to improve data accuracy, identify more reliable data sources and raise awareness of potential bias.

## ACKNOWLEDGEMENTS

We thank Dr. Lorien Dalrymple for her early contribution to the conception of the work.

### Funding

Heejung Bang and Yi Mu were partly supported by Dialysis Clinic, Inc. Heejung Bang was additionally supported by the National Institutes of Health through grant UL1 TR001860. There was no additional external funding received for this study. The funders had no role in study design, data collection and analysis, decision to publish, or preparation of the manuscript.

### Grant Disclosures

The following grant information was disclosed by the authors:
Dialysis Clinic, Inc.
National Institutes of Health through grant:  UL1 TR001860.

### Competing Interests

The authors declare there are no competing interests.

### Author Contributions

- Yi Mu and Heejung Bang conceived and designed the experiments, performed the experiments, analyzed the data, prepared figures and/or tables, authored or reviewed drafts of the paper, approved the final draft.
- Andrew I. Chin conceived and designed the experiments, contributed reagents/materials/analysis tools, authored or reviewed drafts of the paper, approved the final draft, obtaining funding.
- Abhijit V. Kshirsagar performed the experiments, contributed reagents/materials/analysis tools, authored or reviewed drafts of the paper, approved the final draft, expert knowledge and mentoring of the 1st author.

## Data Availability

Sample statistical programs (e.g., in SAS) can be provided upon request from the first author. Data can not be publically shared due to our data use agreement with the data providers. We obtained the data from the US Renal Data System via the National Institute of Diabetes and Digestive and Kidney Diseases (NIDDK) by completing, submitting and following the standard data use agreement. Interested readers should contact the Program Director at the NIDDK (e.g., Dr. Keven Abbott currently) and use https://www.usrds.org/request.aspx.

## Supplemental Information

Supplemental information for this article can be found online at http://dx.doi.org/10.7717/peerj.5284#supplemental-information.

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
