# Peer review of "Data concordance between ESRD Medical Evidence Report and Medicare claims: is there any improvement?"

_PeerJ, doi:10.7717/peerj.5284_

## Round 0.1 · original submission · Major Revisions

Your paper was reviewed by 2 independent experts in this field and they have raised several substantive issues which will need to be thoroughly addressed in any subsequent revisions. In addition, revision of the Tables for clarity, especially Table 3 is highly recommended.

Reviewer 1 ·

Basic reporting

Line 70, lack of continuous validation of which information?
Line 72-73, this sentence needs to be supported by reference or examples of how this data is being used in policy and practice, and as such why the data accuracy in the USRDS matters?
Line 83, explain what does Kappa statistic represents before providing the values.
Line 92, which three methods? Besides, claims data are not considered complicated data source for health services research compared to other data sources from the EMR, such as lab data.
Line 134, it is ZIP code

Experimental design

Line 96, what is the rational for including environmental-related factors, which factors? The authors need a framework or conceptual model for justifying which variables to include and which hypotheses they will test.
The study is a-theoretical, besides there is no clear objectives and hypotheses that the authors are studying. It seems they included variables that they had access to without clear framework or rational.
The authors need to explain in writing in the text the inclusion/exclusion criteria they implemented. Also, in Figure 1, they must include excluded patients and reasons for exclusion.
Why limit Medicare Claims data for past-year, to the minimum, it should include the current year or two or three years before?
What is the value for including the three methods A, B, C. There are well established methods of calculating co-morbidities from claims data and especially for Medicare beneficiaries.

Validity of the findings

145, the authors need to explain what each statistic represents?
149, what is the outcome variable, discordance (0-no/1-yes)? Is it a patient level nested model, which patient level variables did you control for? Also, poverty and education are usually highly correlated, how did you handle the correlation in the regression model? Which model A, B or C did you use in the regression analysis?
Claims data is not considered big data in the current health services research (HSR) terminology, claims data is the customary data that has been used over the past decades to conduct HSR.
Table 3 is unreadable, consider revising the table.

Additional comments

What is the link between the ACA implementation and the analysis, there is nothing in the background that motivates the study and why the ACA implementation would have impacted provider reporting behavior in the CMS form.
The writing quality of the manuscript can be substantially improved and the discussion section needs to be strengthened in light of policy implications and results from previous studies, and provider reporting behavior literature.
Line 251, the authors identified by adding more recent claims data you get more complete picture of co-morbidites (which is very evident), is there a reason why they did not include current year in the analysis, for example government reporting requirement timeline for data? These are details that they need to discuss in the discussion section - how is the values for government measures calculated and what are the implications for practitioners and their behavior?
The study could add value to the literature if the authors address the concerns identified in this and each of the previous sections.

·

Basic reporting

I thought the articulation of study goals, study design and all other aspects of communication specifically in manuscript were all very clear and concise. I'm almost certain the tables/charts would end up in the manuscript as you approach the final draft but it's worth noting that it was a bit distracting the have to open two different documents (or more) if I'm trying to read and review the charts/tables. Lastly, there might be a small typo in the first paragraph (should "form" be "forms"?)

Experimental design

Overall I thought the experimental design was well thought out, organized and clear. Below are some additional thoughts:

1. It might be wise to shed more light on previous results given that the goal of this work is to investigate potential "improvements" vs. pre ACA. Not much discussion takes place around what those old results are and whether or not the differences are meaningful. I'm left having to take your word for it after a little bit of discussion in the "discussion" section. (ex. how does table 1 here compare with the equivalent table in previous studies and is that important to note?)

2. It might also be good to increase the minimum claims requirements for patients to be in the study cohort. CMS made the burden of proof extremely high for co-morbidities being included on claims data for payments. For that reason, you'll notice that a very sparse sample of patients for which any co-morbidities are recorded. That's because providers have to contact hospitals on patients' behalf to collect any evidence of co-morbidities, signed by doctors. Then they have to run that data through a rigorous QC process before any of it shows up on claims. In summary, it might be the case that the ACA had every intention to improve this data capture but, operationally, collecting this data on claims is a nightmare for providers. I acknowledge here that this might be a separate issue worthy of an entirely different public policy effort. But might be helpful to know and control for those issues in your study (i.e. more claims as minimum over a longer time period).

Validity of the findings

The methods, results and conclusions sound robust and accurate given the underlying data/hypothesis. Not much more to add here.

Additional comments

Thanks for giving me the opportunity to review this great work. I'm hopeful it'll drive the right public policy discussion/actions.

---

## Round 0.2 · Minor Revisions

The Reviewer has requested some minor points which can be addressed in the Methods and Results sections and will help to clarify the same for the readers.

Reviewer 1 ·

Basic reporting

The authors have addressed previous comments which substantially improved the quality of the manuscript in this revision. I have few small additional comments.

In which section of the SEER program they use claims data - cases are identified at the center where patients are diagnosed and/or treated from the medical record. Do you mean in the SEER-Medicare dataset?

Page 6, line 19, you mean county?

Did you exclude patients with Medicare Advantage plans from the study?

Table 1, include footnote how you define healthcare utilization and at which level of analysis - State?

Experimental design

No further comments

Validity of the findings

No further comments

Additional comments

No further comments except those in box #1.

---

## Round 0.3 · accepted · Accept

All requested revisions have been addressed adequately. Thank you for the opportunity to review this work.